# Leptospirosis seroprevalence and exposure factors in three informal settlements of French Guiana: An opportunistic survey

Paul Le Turnier[1,2]*, Margot Oberlis[3], Anne Lavergne[4], Loïc Epelboin[1,2], Mathieu Picardeau[5]

**1** Infectious and Tropical Diseases Unit, Guyane University Hospital, Cayenne, French Guiana, **2** UA 17 Santé des Populations en Amazonie, Guyane University Hospital, Cayenne, French Guiana, **3** Mobile Environmental Health Team, French Red Cross, Cayenne, French Guiana, **4** National Reference Center for Hantaviruses - Virology Laboratory, Institut Pasteur of French Guiana, Cayenne, French Guiana, **5** Biology of Spirochetes Unit, National Reference Center for Leptospirosis, WHO Collaborating Center for Reference and Research on Leptospirosis, Institut Pasteur, Paris, France

* paul.leturnier@gmail.com

## Abstract

### Background

Leptospirosis is a zoonotic disease of increasing importance in French Guiana. It particularly affects subjects living in precarious conditions. We aimed to determine the seroprevalence and the risk of exposure to leptospirosis among inhabitants of three informal settlements in French Guiana.

### Methods

A serological investigation was conducted in 2022 in three informal settlements in the area of Cayenne, the main city of French Guiana. Leptospirosis exposure factors were assessed in volunteers aged > 15 through a standardized questionnaire. Leptospirosis seroprevalence was evaluated with Microscopic Agglutination Test (MAT) using 17 pathogenic *Leptospira* antigens with a reactivity threshold of 1:100.

### Results

In 266 participants, median [IQR] age was 42 [34–52] and male to female sex ratio was 0.9. Most participants were migrants (96%), mainly from Haiti (83%), and lived in the study area for at least 2 years (82%). Household rodent exposure (89%) and use of other water sources than collective standpoint (92%) were common. An at-risk occupation was reported for 68% of working participants. Leptospirosis seroprevalence was 7.5% (95% CI [4.7-11.4]) with Ballum and Icterohaemorrhagiae as the main serogroups. Foot skin exposure in wet environments was associated with reactive serum (OR 7.6, 95% CI [1.1 - 326.7]).

which permits unrestricted use, distribution, and reproduction in any medium, provided the original author and source are credited.

**Data availability statement:** The datasets generated and/or analyzed during the current study were made accessible through an agreement with Santé publique France, and this agreement did not include provisions for making the data publicly available. Data processing and initial data collection were conducted under authorization No. 341194v42 granted by the French Data Protection Authority (Commission nationale de l'informatique et des libertés, CNIL) to Santé publique France. However, access to de-identified data may be granted upon reasonable request, after approval from Santé publique France and authorization from CNIL. Requests should be directed to the institutional data access committee via the Direction de la Recherche (DRCI) of Guyane University Hospital (email: drci.promotion@ch-cayenne.fr).

**Funding:** On-field survey "Enquête autour des cas d'Hantavirus, 2022, Santé publique France" was funded by Santé Publique France. Santé Publique France had a role in study design and data collection. The funders had no role in analysis, decision to publish, or preparation of the manuscript. This work was supported by the Agence Nationale de Recherche sur le Sida et les Maladies Infectieuses Emergentes (ANRS/MIE agency) (PLT). ANRS/MIE had no role in study design, data collection and analysis, decision to publish, or preparation of the manuscript. MAT analysis was performed by National Reference Center for Leptospirosis as part of routine analysis.

**Competing interests:** The authors have declared that no competing interests exist.

## Conclusion

Despite a high theoretical risk of leptospirosis exposure among informal settlements inhabitants, only a few participants were seroreactive for *Leptospira*. This may suggest that despite at-risk exposures the effective transmission of leptospirosis remains limited within the study area. Broader serological surveys and environmental studies should clarify the areas of at-risk leptospirosis transmission in French Guiana.

## Author summary

Leptospirosis is a neglected bacterial disease spread mainly through contact with water or soil contaminated by rodents which can cause severe illness. In French Guiana, people living in informal settlements often face precarious conditions such as basic housing, limited access to safe water, and frequent contact with flooded or muddy environments. These circumstances may increase their risk of infection. This serological survey provides the first assessment of leptospirosis exposure in such communities of French Guiana. Residents from three settlements answered questions about their daily lives and provided blood samples. Most were migrants, particularly from Haiti, who had lived in the settlements for several years. Risky situations were very common: rodent presence around households, use of alternative water sources like rainwater or wells, and at-risk occupations. Yet only 7.5% of participants showed signs of past infection using a gold standard serological assay. The only factor associated with previous leptospirosis was walking barefoot or with unprotected feet in wet environments. These results suggest that despite widespread risk factors, actual transmission in the surveyed settlements may be lower than expected. Broader studies, including environmental and rodent investigations, are needed to better identify hotspots of transmission and guide prevention efforts for vulnerable populations.

## Introduction

Leptospirosis is a worldwide bacterial zoonotic disease causing around 1,000,000 cases and 60,000 deaths each year [1]. Leptospirosis has long been considered as a disease of poverty and precariousness because of the association between low incomes, poor sanitation or housing and leptospirosis past exposure assessed by serological survey or leptospirosis incidence as demonstrated in numerous studies especially conducted in Brazilian urban slums [2–4]. French Guiana (FG) is a French overseas territory located on the northeastern coast of South America bordering Brazil and Suriname. French Guiana has a population of approximately 300,000 inhabitants, most of whom live in urban or suburban areas along the Atlantic coast. Over the past decade, demographic growth in informal settlements has been particularly increasing around Cayenne, the main city and could concern up to 20% of inhabitants [5,6]. This phenomenon occurs in a context of growing precariousness for

many inhabitants, as well as continuous migratory flow from the neighboring countries but also from more distant countries such as Haiti, Syria or Afghanistan. A large part of the French Guianese population lives in a high level of precarity across multiple dimensions: social, financial, residential but also sanitary [7–9]. This has serious implications for health, including the local epidemiology of leptospirosis. Between 2016 and 2022, the annual incidence more than doubled compared with 2007–2014, highlighting an increased disease burden [10]. Approximately one third of the affected individuals lacked health insurance, and a similar proportion lived in informal settlements—both figures higher than those observed in the general population. This overrepresentation of uninsured individuals and residents of informal settlements among leptospirosis cases suggests that the disease may disproportionately affect socially and economically vulnerable populations in French Guiana. Poor living conditions in urban informal settlements, particularly limited access to safe water and adequate sanitation, likely contribute to this increased risk. With few exceptions, residents living in informal settlements generally do not have access to running water for drinking, food preparation, and other domestic uses. In these areas, inadequate wastewater and solid waste management leads to environmental contamination, including rudimentary disposal sites, insufficient garbage collection, and high rat populations. These factors provide a favorable setting for the emergence and transmission of leptospirosis. However, few data are available on the population living in these areas, apart from studies conducted by organizations involved in public health prevention, such as the Red Cross [11,12]. Especially, no data exist in French Guiana on leptospirosis exposure among people living in informal settlements. Collecting local data, whether through risk assessment or serological surveys, is a necessary first step to guide public health strategies in these communities.

The primary objective of the study was to assess the prevalence of prior *Leptospira* exposure in a population living in informal settlements near Cayenne using MAT serological analysis, and to investigate factors associated with seropositivity. The secondary objective was to describe risk factors for leptospirosis exposure among residents of informal settlements in French Guiana.

## Materials and methods

### Ethics statement

All included subjects gave their written informed consent before being enrolled in the survey and accepted to participate in this ancillary study. Health mediators helped participants fill out the questionnaire when necessary. The implementation of data processing and initial data collection were conducted in accordance with authorization No. 341194v42 granted by the French Data Protection Authority (Commission nationale de l'informatique et des libertés, CNIL) to Santé Publique France (the French Public Health Agency). All analyses were performed on anonymized datasets, and procedures adhered to the ethical standards of the National Research Committee and the Declaration of Helsinki.

As part of the collaboration, a confidentiality agreement was signed between the authors and Santé Publique France to ensure compliance with specific data protection provisions, including the restriction of reporting results based on very small sample sizes that could pose an identification risk. Accordingly, no results were reported for units with fewer than five participants, and data were aggregated when necessary.

### On-field investigation and sampling

In 2022, an unusual number of cases of Hantavirus-associated pulmonary syndrome (HPS) was reported to health authorities of French Guiana.11 Four cases were diagnosed over a 7-month period while only 7 cases had occurred during the 2008 – 2021 period.12. This was considered as a possible severe emergence of the disease and led to conduct investigations around cases. After determining the place of living and probable place of contamination, three areas adjacent to Cayenne were selected for further on-site investigations. These sites were Boutilier in the municipality of Remire-Montjoly and an area named PK13 for kilometric point 13 and PK 16 for kilometric point 16 two sites in the

municipality of Macouria. Multidisciplinary teams composed of health mediators, physicians specialized in infectious and tropical diseases and epidemiology and nurses were set up. The precise methodology used in this investigation around HPS cases has already been reported [12]. Awareness-raising sessions were conducted prior to the serological survey and on the days of on-site investigations by health mediators. Inhabitants were asked to participate during outreach visits. The inclusion criteria were age over 15 and residing in the study area (without duration limit). Participants were excluded if they declined participation in any ancillary study beyond the Hantavirus survey. The survey took place from November 28th to December 16th, 2022. For a same site, sessions of investigation were repeated on different days to enable greater participation from local eligible inhabitants. After providing information and obtaining written consent from participants, a questionnaire was administered. French and Haitian Creole were the main languages used for survey administration. Guyanese Creole, English-speaking Guyanese Creole, Portuguese, Spanish and English were also used to a lesser extent. Information on the risk of rodent-transmitted diseases and protective measures as well as a hygiene kit were provided after performing the survey for each participant. Finally, several weeks after the initial survey, a campaign of results delivery was organized. Participants were approached individually by awareness-raising sessions but also through collective information campaigns at each of the three sites investigated. The Hantavirus serology results were communicated individually from 23rd of January to 27th of January.

### Data collection methods and definitions

The on-site initial Hantavirus survey included a standardized questionnaire as well as serum sampling to investigate immunization against Hantavirus and possibly further serological tests.

The initial questionnaire comprised sociodemographics data, housing conditions and exposure factors to Hantavirus infection. (see supplementary data S1 Text).

Before the results were provided, an ancillary study was designed to investigate supplementary leptospirosis exposure factors among the survey participants (see supplementary data S2 Text). Thus, a complementary questionnaire was designed and administered to any volunteer encountered during the delivery of results of Hantavirus serology.

Occupations considered at-risk due to frequent contact with animal-soiled or moist environments included: agriculture, maintenance of urban green spaces, gardening, odd jobs, housekeeping/ cleaning, construction and public works, fishing. Non-at-risk occupations were: security guard and logistics, personal assistance, commerce, student, auto mechanic, office worker.

### Microscopic agglutination test

After on-field sampling, all sera were refrigerated during transport to Institut Pasteur de Guyane, Cayenne, French Guiana. They were centrifuged then stored at -80°C until further analyses. Five hundred µl of serum was tested to determine its reactivity. A previous contact with *Leptospira* sp. was defined as the presence of a reactive serum. It was initially anticipated to use a positivity threshold of 1:50 for the Microscopic Agglutination Test (MAT). However, because many serum samples were contaminated, agglutination patterns were difficult to interpret at this dilution. The MAT was therefore performed starting at a 1:100 dilution, which reduced background interference and improved the readability of results. Finally, sera with MAT titer ≥1:100 were considered reactive, while those <1:100 were non-reactive, as previously reported [13–15]. MAT analyses were all performed at the same time at the French National Reference Center of Leptospirosis (NRCL), Institut Pasteur, Paris, France in June 2025. The choice of the panel of 17 *Leptospira* strains for MAT analysis was based on previous reports from French Guiana [16] and the included serogroups were Australis, Ballum, Bataviae, Canicola, Celledoni, Cynopteri, Djasiman, Grippotyphosa, Hebdomadis, Icterohaemorrhagiae, Javanica, Mini, Panama, Pomona, Sarmin, Sejroë, and Tarassovi. In case of reactive serum, the determination of presumptive infective serogroup was based on the highest observed MAT titer.

## Statistical analysis

Quantitative variables were described with mean and standard deviation or median and interquartile range, depending on the variable distribution, and qualitative variables were described with frequency and percentage. Confidence intervals (95% CI) were estimated for proportions.

The association of variables with reactive serum was tested in bivariate analysis. Proportions were compared with Chi-square test and Fisher's exact test as appropriate. Crude odds ratio (OR) were determined with 95% CI. Wilcoxon rank-sum test was performed for quantitative variables. We tested the hypothesis that accumulation of exposure factors may be associated with a reactive serum. Thus, a cumulative score was computed among the subgroup of participants in whom the complementary questionnaire was administered. The cumulative score was based on the following exposure factors: exposure to water and soil, water sources and management, rodent exposure, occupational and outdoor activities, housing and environmental conditions. The 17 variables used for computing the cumulative exposure score are given in supplementary data (see supplementary data S1 Table).

All statistical tests were two-sided and a p-value of 0.05 was considered significant. Missing data were not considered in the calculation of OR and p-values.

Analyses were performed using the STATA Now/SE 19.5 software (College Station, TX: StataCorp LLC).

## Results

The initial Hantavirus survey was conducted in 274 volunteers, which corresponded to roughly 30% of the inhabitants in study areas based on previous estimations [12]. Among them, 266 (97.1%) accepted to participate in this ancillary study. Finally, investigators were able to administer the complementary questionnaire to 190/266 (71.4%) participants during individual Hantavirus results delivery.

Fig 1 details the flow chart of the reported leptospirosis ancillary survey.

Median age of participants was 42 (34.4-51.7). Age distribution is presented in supplementary S1 Fig. Women accounted for 54.0% (143/265) of cases. Participants born in mainland France or French Guiana accounted for 3.9% (9/233) of participants. Participants were born mostly in Haiti (82.8%, 193/233). Sociodemographic data are given in Table 1. Participants born abroad had been living in French Guiana for more than two years before the survey took place in 92.6% (174/188) of cases. Participants resided in the study area for less than one year in 7% (13/186) of cases and more than 1, 2 and 5 years for 93.0% (173/186), 81.7% (152/186) and 43.6% (81/186) of them respectively. Distribution of the duration of living in French Guiana (supplementary data S2 Fig) and in the study area (supplementary data S3 Fig) are provided.

Half (54.5%, 145/266) participants lived in a *carbet* defined by informal housing made of metal sheets and wood. However, 83.1% of participants (246/264) had tiling or concrete as primary flooring material. Housing conditions are summarized in Table 1.

With regard to domestic freshwater use, most participants frequently used the prepaid water standpipe in their neighborhood (78.6% 209/266).

More than 90% of respondents to the complementary survey used an alternative source in addition to the neighborhood collective water distribution standpipe. Among them, nearly half used a well or collected rainwater. Half of the well users did not use protection from flooding. Two thirds of the users of rain water collector protected it with a lid. Table 2 details the water collection sources and usages of participants.

Over half (54.9% 146/266) reported a professional activity. Construction worker (30.8%), odd jobs (15.8%), housekeeping/ cleaning (14.4%) and commerce (13.0%) were the most frequent occupations. Table 3 details occupational and non-occupation individual exposures. Among, those reporting professional activity, 67.8% (99/146) were estimated to have an occupation at risk for leptospirosis.

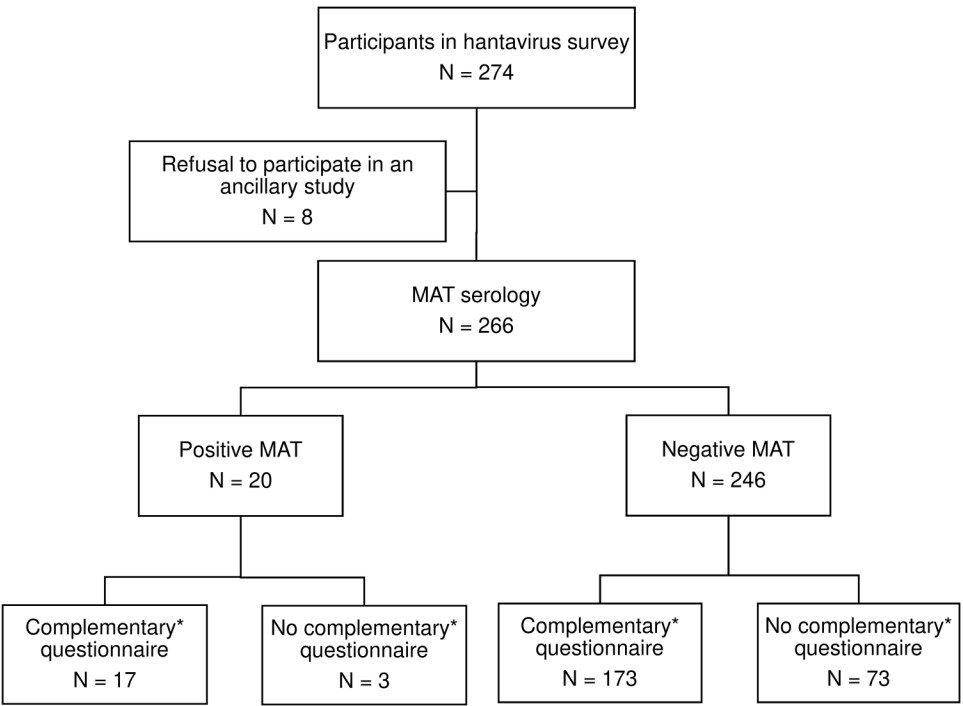

**Fig 1. Flow chart of participants in leptospirosis serological survey.** MAT Microscopic agglutination test. *Complementary questionnaire was administered during the delivery of Hantavirus serology results to 190 participants.

Overall the median (IQR, range) number of cumulative risk factors investigated by the questionnaires was 7 (6–8; 2–12), considering only the participants who responded to the ancillary survey (n = 190).

MAT analysis revealed that sera were reactive for 7.5% 95% CI [4.7-11.4] (20/266) participants. Among the 20 reactive sera, 5 presumptive infective serogroups were identified: Ballum (n = 9), Icterohaemorrhagiae (n = 6), Panama (n = 3), Australis (n = 1) and Tarassovi (n = 1), with a median [IQR] titer of 100 [100–200]. Results of MAT analysis are presented in Table 4.

Bivariate analysis revealed that walking barefoot or with open shoes during heavy rainfall or in stagnant water was more common among seroreactive participants (93.3%, 14/15) than non-reactive participants (64.9%, 96/148), with an odds ratio of 7.6 (95% CI: 1.1–326.7, p = 0.039).

The median (IQR, range) number of cumulative risk factors was 7 (6–9; 5–11) and 7 (6–8; 2–12) for seroreactive and non-reactive participants, respectively (p = 0.491). Logistic regression analysis showed that an increase of the number of risk factors was not associated with a higher risk of positive MAT test, with an OR of 1.12 (95% CI: 0.86–1.45) per one additional risk factor (see Table 3).

## Discussion

Leptospirosis at-risk exposures were very common among surveyed participants of three informal settlements of French Guiana, especially rodent vicinity. This proximity with rodent combined with housing type characteristics, appears to be a relevant criterion to estimate the risk of severe leptospirosis. A study conducted in urban slum populations in Salvador, Brazil, revealed that signs of rodent infestation close to the household as well as unplastered walls were risk factors for the incidence of severe leptospirosis [17]. However, the magnitude of association between rodent proximity, rat abundance or *Leptospira* excretion rate by rats and *Leptospira* seroprevalence has been debated. In fact, environmental

**PLOS Neglected Tropical Diseases**

**Table 1. Sociodemographic data and residence characteristics.**

| | All participants<br>N = 266 | 95% CI | Reactive MAT<br>N = 20 | Non - reactive MAT<br>N = 246 | OR | 95% CI | p - value |
|---|---|---|---|---|---|---|---|
| **Age**, years | 42.0 (34.4 - 51.7) | | 42.0 (34.5 - 51.8) | 41.9 (31.5 - 48.3) | NA | NA | 0.57 |
| **Male gender** | | | | | | | |
| - Yes | 122 (46.0) | 39.9 - 52.2 | 9/20 (45.0) | 113/245 (46.1) | 0.96 | 0.34 – 2.64 | 0.9229 |
| - No | 143 (54.0) | 48.8 - 60.1 | 11/20 (55.0) | 132/245 (53.9) | | | |
| - Missing | 1 | | 0 | 1 | | | |
| **Country of birth** | | | | | | | |
| - Haiti | 193 (82.8) | 77.4 - 87.4 | 11/13 (84.6) | 182/220 (82.7) | – | – | – |
| - Guyana | 15 (6.4) | 3.6 - 10.4 | 1/13 (7.7) | 14/220 (6.4) | | | |
| - France | 9 (3.9) | 1.8 - 7.2 | 0 | 9/220 (4.1) | | | |
| - Brazil | 8 (3.4) | 1.5 - 6.7 | 1/13 (7.7) | 7/220 (3.2) | | | |
| - Other | 8 (3.4) | 1.5 - 6.7 | 0 | 8/220 (3.6) | | | |
| - Missing | 33 | – | 7 | 26 | | | |
| **Country of birth,** | | | | | | | |
| - Haiti | 193 (82.8) | 77.4 - 87.4 | 11/13 (84.6) | 182/220 (82.7) | 1.15 | .24 - 11.1 | 0.86 |
| - Different from Haiti (ref.) | 40 (17.2) | 12.6 - 22.6 | 2/13 (15.4) | 38/220 (17.3) | | | |
| - Missing | 33 | | 7 | 26 | | | |
| **Duration of living in French Guiana\*,\*\*,** | | | | | | | |
| - years | 6.4 [4.3 - 9.1] | – | 6.2 (4.0 - 11.8) | 6.4 (4.3 - 8.9) | – | – | 0.69 |
| - >1 y | 185 (98.4) | 95.4 - 99.7 | 17/17 (100) | 168/171 (98.3) | | | |
| - >2 y | 174 (92.6) | 87.8 - 95.9 | 15/17 (88.2) | 159/171 (93.0) | | | |
| - >5 y | 127 (65.5) | 60.4 - 74.2 | 12/17 (70.3) | 115/171 (67.3) | | | |
| - >10 y | 44 (23.4) | 17.6 - 30.1 | 5/17 (29.4) | 39/171 (22.8) | | | |
| - >20 y | 33 (17.6%) | 12.4 - 23.8 | 3/17 (17.7) | 30/171 (17.5) | | | |
| - Missing | 78 | | 3 | 75 | | | |
| **Duration of living in the study area\*** | | | | | | | |
| - years | 4.5 [2.3 - 6.6] | – | 3.7 [4.4 - 6.1] | 4.6 [2.7 - 6.6] | – | – | 0.38 |
| - <1y | 13 (7.0) | 3.8 - 11.7 | 0 | 13/169 (7.7) | | | |
| - >1 y | 173 (93.0) | 88.3 - 96.2 | 17/17 (100) | 156/169 (93.0) | | | |
| - >2 y | 152 (81.7) | 75.4 - 87.0 | 13/17 (76.5) | 139/169 (82.3) | | | |
| - >5 y | 81 (43.6) | 36.3 - 51.0 | 5/17 (29.4) | 76/169 (13.6) | | | |
| - >10 y | 26 (14.0) | 9.3 - 19.8 | 3/17 (17.7) | 23/169 (13.6) | | | |
| - >20 y | 11 (5.9) | 3.0 - 10.3 | 0 | 11/169 (6.5) | | | |
| - Missing | 80 | | 3 | 77 | | | |
| **Place of residence (study area)** | | | | | | | |
| - *PK 16* | 159 (59.8) | 53.6 - 65.7 | 15/20 (75.0) | 144/246 (58.5) | NA | NA | NA |
| - *Boutilier* | 91 (34.2) | 28.5 - 40.3 | 5/20 (25.0) | 86/246 (35.0) | | | |
| - *PK 13* | 16 (6.0) | 3.5 - 9.6 | 0 (0) | 16/246 (6.5) | | | |
| **Housing type** | | | | | | | |
| - House – concrete, cement or cinderblock (ref.) | 117 (44.0) | 48.3 - 60.6 | 10/20 (50.0) | 107/246 (43.5) | 0.77 | 0.28 – 2.15 | 0.5731 |
| - *Carbet* – metal sheets and wood or other | 149 (56.0) | 59.8 - 62.1 | 10/20 (50.0) | 139/246 (56.5) | | | |

*(Continued)*

**Table 1.** (Continued)

| | All participants | 95% CI | Reactive MAT | Non - reactive MAT | OR | 95% CI | p - value |
|---|---|---|---|---|---|---|---|
| | N = 266 | | N = 20 | N = 246 | | | |
| **Primary flooring material** | | | | | | | |
| - Tiling or concrete (ref.) | 246 (93.2) | 89.4 - 95.9 | 19/20 (95.0) | 227/244 (93.0) | 0.7 | 0.09 - 5.57 | 0.738 |
| - Dirt/wooden/metallic | 18 (6.8) | 4.1 - 10.6 | 1/20 (5.0) | 17/244 (7.0) | | | |
| - Missing | 2 | | 0 | 2 | | | |

Data are median [IQR] for quantitative variable and n (%) for categorical variable.

For ancillary questions, the number of participants responding (n = X) is shown and n/N is given for the comparison of reactive vs non - reactive.

*Variable could be collected only in participants who answered the complementary questionnaire, explaining a high number of missing data.

** this question was asked only for those who were born abroad (n = 224).

CI Confidence Interval, MAT Microscopic agglutination test, OR crude Odds ratio.

factors such as inadequate source of water, sewage, flooding risk and sanitation infrastructure as well as other environmental characteristics have been reported as even more important drivers for leptospirosis transmission [18,19]. Nevertheless, *Rattus* spp. remains the main reservoir of *Leptospira* and is involved in the transmission cycle of human leptospirosis and rat exposure has been associated with *Leptospira* seroprevalence at various degrees [20–22]. The only available data on the rate of *Leptospira* excretion by invasive rodents in French Guiana are outdated and provide little informative value. [23]. In Brazil, reports have provided rates of *Leptospira* excretion by rats of 31–92% [24]. Measuring the prevalence of *Leptospira* excretion in locally collected rats could provide valuable data to better assess environmental circulation in the area. Most participants frequently used a source other than the collective prepaid water standpipe. While it seems reasonable to assume that the standpipe-collected water provides free-from-*Leptospira* water, a regular use of the standpipe may not be totally harmless [25]. In fact, residents often go to the standpipe wearing open shoes or even barefoot and can be in contact with contaminated soil. Indeed, standpipes are often located at the bottom of residential areas where water can drain or accumulate and increase the risk of leptospirosis [22,26,27]. In addition, standpipes are also often close to waste deposits at the entry of the settlements (see supplementary S4 Fig). Regarding the use of unsafe water, different degrees of risk may exist depending on the type and provenance of water source used [28]. The use of wells or rainwater collection were reported by half of the participants whereas they are both risk factors for leptospirosis especially when not protected [14,29–31]. Providing information and protective measures to protect the collected water from contamination should be a priority and be considered as an actionable measure to decrease the risk of leptospirosis in informal settlements [32].

A previous leptospirosis infection was found in 7.5% (95% CI [4.7-11.4]) of patients, a lower rate than expected given the high level of exposure. In a study conducted in Cali, Colombia, seroprevalence (MAT, titer threshold 1: 100) was 12% (95% CI [10–14]), slightly higher than the present study despite most of participants had access to running water [33]. In another study conducted in an urban community of Puerto Rico, a higher seroprevalence (MAT, titer threshold 1: 50) of 27.2 (95% CI [21.2-33.9]) was found [34]. Delight et al. recently reported a large study conducted among urban slum residents of Salvador, Brazil, and found a seroprevalence (MAT, titer threshold 1: 100) of 11.3% (95% CI [9.1–13.8]) again slightly higher but in the same magnitude as the reported study [35]. Many factors were associated with seropositivity with a differential effect depending on gender (see below). These three seroprevalence studies were conducted in response to occurrence of cases locally [33,35] or because environment was favorable for leptospirosis transmission as done in French Guiana [34].

So far, the unique study of human leptospirosis seroprevalence conducted in French Guiana has been performed among illegal goldminers operating in FG and investigated mostly men (73%). The seropositivity (MAT, titer threshold

**Table 2. Water collection sources and usages.**

| | All participants N = 266 | 95% CI | Reactive MAT N = 20 | Non - reactive MAT N = 246 | OR | 95% CI | p - value |
|---|---|---|---|---|---|---|---|
| **Common access to the neighborhood prepaid water standpipe** | | | | | | | |
| -Yes | 209 (78.6) | 73.1 - 83.3 | 16/20 (80.0) | 193/246 (78.5) | 1.1 | 0.33 - 4.70 | 1 |
| -No | 57 (21.4) | 16.7 - 26.9 | 4/20 (20.0) | 53/246 (21.5) | | | |
| **• If no (n = 57), main alternate source of water collection** | | | | | | | |
| -Well | 15/40 (37.5) | | 0 | 15/36 (40.5) | 1.1 | 0.33 - 4.70 | 1 |
| -Household water tap | 9/40 (22.5) | | 0 | 9/36 (21.7) | | | |
| -Rainwater collection | 9/40 (22.5) | | 4/4 (100) | 5/36 (18.9) | | | |
| -Commercially bottled water | 5/40 (12.5) | | 0 | 5/37 (13.5) | | | |
| -Other | 2/40 (5.0) | | 0 | 2/36 (5.4) | | | |
| Missing | 17 | | | | | | |
| **Use of other source than the prepaid water standpipe *** | | | | | | | |
| -Yes | 174 (91.6) | 86.7 - 95.1 | 17/17 (100) | 157/173 (90.8) | – | – | 0.37 |
| -No (ref.) | 16 (8.4) | 4.9 - 13.3 | 0 | 16/173 (9.2) | | | |
| -Missing | 76 | | 3 | 73 | | | |
| **• If yes, type of alternative source (n = 174)** | | | | | | | |
| -Well | | | | | | | |
| ◦Yes | 94 (54.7) | 46.9 - 62.2 | 10/17 (58.8) | 84/155 (54.2) | 1.21 | 0.39 - 3.94 | 0.80 |
| ◦No (ref.) | 78 (45.3) | 37.8 - 53.1 | 7/17 (41.2) | 71/155 (45.8) | | | |
| ◦Missing | | | | | | | |
| -Rainfall | | | | | | | |
| ◦Yes | 2 | | 0 | 2 | 1.30 | 0.42 - 4.08 | 0.62 |
| ◦No (ref.) | 81 (47.1) | 39.4 - 54.8 | 9/17 (52.9) | 72/155 (46.5) | | | |
| ◦Missing | | | | | | | |
| **• Retention basin connected to a piped supply system** | | | | | | | |
| ◦Yes | 91 (52.9) | 45.1 - 60.5 | 8/17 (47.1) | 83/155 (53.5) | 0.73 | 0.13 - 2.85 | 0.77 |
| ◦No (ref.) | 2 | | 0 | 2 | | | |
| ◦Missing | | | | | | | |
| **• Creek/watercourse** | | | | | | | |
| ◦Yes | 38 (22.1) | 16.1 - 39.0 | 3/17 (17.7) | 35/155 (22.6) | 0.53 | 0.06 - 2.50 | 0.53 |
| ◦No (ref.) | 134 (77.9) | 70.9 - 83.9 | 14/17 (82.3) | 120/155 (77.4) | | | |
| ◦Missing | 2 | | 0 | 2 | | | |
| **• Use of a lid to cover the alternative water source (n = 174)** | | | | | | | |
| -Yes | 93 (66.0) | 57.5 - 73.7 | 12/14 (85.7) | 81/127 (63.8) | 3.41 | | 0.14 |
| -No (ref.) | 48 (34.0) | 26.3 - 46.5 | 2/14 (14.3) | 46/127 (36.2) | | | |
| Missing | 33 | | 3 | 30 | | | |
| **• Flood protection practices among well users (n = 94)** | | | | | | | |
| -Yes | 44 (52.4) | 33.3 - 50.1 | 3/10 (30.0) | 41/74 (55.4) | 0.34 | | 0.18 |
| -No (ref.) | 40 (47.6) | 41.2 - 63.4 | 7/10 (70.0) | 33/74 (44.6) | | | |
| -Missing | 10 | | 0 | 10 | | | |

Data are n (%). For ancillary questions, the number of participants responding (n = X) is shown and n/N is given for the comparison of reactive vs non - reactive. Fisher's exact test was used for comparison.

CI Confidence Interval, MAT Microscopic agglutination test, OR crude Odds ratio, ref. Reference.

*Variable was collected only in participants who answered the complementary questionnaire, explaining a high number of missing data.

PLOS Neglected Tropical Diseases

**Table 3. Occupational and other individual exposure factors.**

| | All participants N=266 | 95% CI | Reactive MAT N=20 | Non - reactive MAT N=246 | OR | 95% CI | p - value |
|---|---|---|---|---|---|---|---|
| **Occupation** | | | | | | | |
| - Yes | 146/266 (54.9) | 48.7 - 61.0 | 11/20 (55.0) | 135/246 (54.9) | 1 | 0.36 - 2.85 | 0.992 |
| - No (ref.) | 120/266 (45.1) | 39.0 - 51.3 | 9/20 (45.0) | 111/246 (45.1) | | | |
| **• Occupation category (n=146)** | | | | | | | |
| - Construction worker | 45 (30.8) | | 5/11 (45.5) | 40 (29.6) | NA | NA | NA |
| - Odd jobs/ Temporary jobs | 23 (15.8) | | 0 | 23 (17.0) | | | |
| - Housekeeping/ Cleaning | 21 (14.4) | | 2/11 (18.2) | 19 (14.1) | | | |
| - Trade/ Commerce | 19 (13.0) | | 02-Nov | 17 (12.6) | | | |
| - Student | 6 (4.1) | | 01-Nov | 5 (3.7) | | | |
| - Logistics and security | 6 (4.1) | | 0 | 6 (4.4) | | | |
| - Agriculture | 5 (3.4) | | 0 | 5 (3.7) | | | |
| - Auto mechanic | 5 (3.4) | | 0 | 5 (3.7) | | | |
| - Urban green spaces maintenance | 5 (3.4) | | 1 (9.1) | 4 (3.0) | | | |
| - Personal assistance | 5 (3.4) | | 0 | 5 (3.7) | | | |
| - Other | 6 (4.1) | | 0 | 6 (4.4) | | | |
| **• At - risk occupation (n=146)** | | | | | | | |
| - Yes | 99 (67.8) | 59.6 - 75.3 | 8/11 (72.7) | 91/135 (67.4) | 1.29 | 0.29 – 7.90 | 1 |
| - No (ref.) | 47 (32.2) | 24.7 - 40.4 | 3/11 (27.3) | 44/135 (32.6) | | | |
| **Agricultural or forestry activities in contact with the ground** | | | | | | | |
| - Yes | 70 (26.3) | 21.1 - 32.0 | 7/20 (35.0) | 63/246 (25.6) | 1.56 | 0.50 – 4.44 | 0.43 |
| - No (ref.) | 196 (73.7) | 68.0 - 78.9 | 13/20 (65.0) | 183/246 (74.4) | | | |
| **Waste near household** | | | | | | | |
| - Yes | 65 (24.4) | 19.4 - 30.1 | 3/20 (15.0) | 62/246 (25.2) | 0.52 | 0.10 – 1.90 | 0.42 |
| - No (ref.) | 201 (75.6) | 69.9 - 80.6 | 17/20 (85.0) | 184/246 (74.8) | | | |
| **Household use of rodent traps or rodenticide** | | | | | | | |
| - Yes | 127 (48.1) | 41.9 - 54.3 | 6/19 (31.6) | 121/245 (49.4) | 0.47 | 0.14 – 1.39 | 0.1345 |
| - No (ref.) | 137 (51.9) | 45.7 - 58.1 | 13/19 (68.4) | 124/245 (50.6) | | | |
| - Missing | 2 | | 1 | 1 | | | |
| **Handling rodent (dead or alive)** | | | | | | | |
| - Yes | 20 (7.6) | 4.7 - 11.4 | 3/19 (15.8) | 17/246 (7.0) | 2.53 | 0.43 – 10.13 | 0.163 |
| - No (ref.) | 245 (92.4) | 88.6 - 95.3 | 16/19 (84.2) | 229/246 (93.0) | | | |
| - Missing | 1 | | 1 | | | | |
| **Rodent signs in/around household****** | | | | | | | |
| - Yes | 169 (89.0) | 83.6 - 93.0 | 16/17 (94.1) | 153/173 (88.4) | | 0.29 – 92.02 | 0.699 |
| - No (ref.) | 21 (11.0) | 07.0 - 16.4 | 1/17 (5.9) | 20/173 (11.6) | 2.09 | | |
| - Missing | 76 | | 3 | 73 | | | |
| **Frequent skin wounds (hands or feet)****** | | | | | | | |
| - Yes | 52 (28.0) | 21.6 - 35.0 | 4/17 (23.5) | 48/169 (28.4) | 0.78 | 0.18 – 2.68 | 0.783 |
| - No (ref.) | 134 (72.0) | 65.0 - 78.4 | 13/17 (76.5) | 121/169 (71.6) | | | |
| - Missing | 80 | | 3 | 77 | | | |

*(Continued)*

**Table 3.** (Continued)

| | All participants N = 266 | 95% CI | Reactive MAT N = 20 | Non - reactive MAT N = 246 | OR | 95% CI | p - value |
|---|---|---|---|---|---|---|---|
| **Walking barefoot or in open shoes around household**\*\* | | | | | | | |
| - Yes | 163 (85.8) | 80.0 - 90.4 | 15/17 (88.2) | 148/173 (85.5) | 1.27 | 0.27 – 12.08 | 1 |
| No (ref.) | 27 (14.2) | 9.6 - 20.0 | 2/17 (11.8) | 25/173 (14.5) | | | |
| **• If yes, also during heavy rainfall or in stagnant water (n = 163)** | | | | | | | |
| - Yes | 110/163 (67.5) | 59.7 - 74.6 | 14/15 (93.3) | 96/148 (64.9) | 7.58 | 1.09 – 326.65 | 0.039 |
| - No (ref.) | 53/163 (32.5) | 25.4 - 40.3 | 1/15 (6.7) | 52/148 (35.1) | | | |

Data are n (%). N provides participants with available data and n/N is given for the comparison of reactive vs non - reactive. Chi - square test or Fisher's exact test were used as appropriate.

CI Confidence Interval, MAT Microscopic agglutination test, OR crude Odds ratio.

\*\*Variable collected through the complementary questionnaire only administered to 190 participants, explaining a high number of missing data.

1: 50) rate was notably higher than in the present study population in 2015 (31.0%, ORPAL1 study) and 2019 (28.1%, ORPAL2 study). Since a different titer threshold was used, comparison may be questionable. Thus, to better contextualize our findings, we reanalyzed the ORPAL1 and ORPAL2 data on MAT seroprevalence in collaboration with the ORPAL study investigators (Prof. Maylis Douine, personal communication). Applying a 1:100 cutoff, the seropositivity rate was 19.2% and 16.2% for 2015 and 2019 respectively. Both were still significantly higher than the seropositivity rate observed in informal settlements. A hypothesis to explain this higher seroprevalence would be a high level of mud exposure during gold mining activity associated with a high level of environmental contamination of gold mining sites. A contamination of illegal goldminers outside of gold mining sites is also a credible situation given the precariousness of this population [36].

The moderately elevated seropositivity rate observed in the study may suggest two hypotheses. First, if we consider that there is a high level of environmental contamination it should imply that preventive measures (not investigated) are taken to avoid the penetration of the bacteria. The role of individual preventive measure has already been reported [29]. This would imply that the supposed exposure factors, which were frequently reported and cumulated among informal settlements inhabitants may not actually be as risky as expected in this context. On the contrary, if we acknowledge that the level of cumulative exposure factors was indeed elevated, this could imply that the environment of study sites may possibly be less contaminated than anticipated. Repeating surveys in other settings could help to explore these hypotheses as well as exploring environmental contamination through soil and surface water analysis.

The sole factor associated with MAT seropositivity in the reported study was a feet skin exposure to wet environments during heavy rain or in stagnant water. Although the confidence interval is wide, this factor appears plausible given known leptospirosis transmission pathways and the aforementioned risk of transmission around standpipes. In contrast, male gender was the only factor associated with seropositivity among illegal goldminers working in French Guiana. A recent study conducted across four informal settlements at-risk communities in Salvador, Brazil used an original gender-based analysis to investigate differential effects on risk perception, exposure, and at-risk activities for leptospirosis [35]. Male gender and increasing age were the unique factors related to positive *Leptospira* serostatus when investigating the whole study population. Interestingly, men who perceived leptospirosis as extremely serious had lower odds of being seropositive and engage in at-risk activity such as walking through sewage and walking barefoot outside. The impact of diseases severity perception was different among men and women. Besides, at-risk occupation increased the risk for seropositivity in men while age increased the seropositivity rate among women. This suggests a clear differential effect of gender on the modalities of leptospirosis exposure.

This study has several limitations, particularly the fact that many patients did not answer to the second questionnaire generating missing data and its non-random sampling design. All participants were recruited after being sensitized through

**Table 4. Microscopic agglutination test results.**

| Presumptive serogroup identified by MAT | Participants with positive MAT N (%) | Titer distribution |
|---|---|---|
| Ballum | 9 (45) | 100 (n = 7) 200 (n = 2) |
| Icterohaemorrhagiae | 6 (30) | 100 (n = 5) 200 (n = 1) |
| Panama | 3 (15) | 100 (n = 2) 200 (n = 1) |
| Australis | 1 (5) | 100 (n = 1) |
| Tarassovi | 1 (5) | 1600 (n = 1) |
| Bataviae, Canicola, Celledoni, Cynopteri, Djasiman, Grippotyphosa, Hebdomadis, Javanica, Mini, Pomona, Sarmin, Sejröe | 0 | – |

MAT Microscopic agglutination test, % are given with total positive results as denominator.

outreach campaigns, either several days before sampling or on the same day just prior to participation. Consequently, we cannot exclude the possibility that this pre-intervention sensitization about a rodent-borne disease (HPS) may have biased some of the questionnaire responses. Nevertheless, more than 90% of sensitized inhabitants expressed their intention to participate, and nearly 80% eventually did so [12]. The attrition between sensitization and sampling was therefore limited, suggesting that inclusion bias among the sensitized population was minimal. The main limitation relates to representativeness and generalizability, since the outreach campaigns reached only about one-third of the estimated population across the three sites, despite the investigators' extensive efforts. To better estimate the seroprevalence of leptospirosis and associated factors across French Guiana, larger and random sampling should be considered as performed for other zoonotic diseases [37,38]. Additionally, the time and place of acquisition of the previous exposure to leptospirosis cannot be ascertained since MAT remains positive for several years [39]. Unfortunately there is no data regarding the epidemiology of leptospirosis in Haiti to discuss the seroprevalence of the participants born in Haiti. Since 1965 the sole published study of leptospirosis cases from Haiti is a case series of three patients who needed air medical transport [40]. Most people born in Haiti who migrate to French Guiana travel through Brazil or Suriname. There are no published data on the epidemiology of leptospirosis in Suriname. In Brazil, numerous studies have investigated leptospirosis across various regions and contexts. However, the specific regions and duration of transit for each migrant participant were not assessed. For these reasons, further analyses on this aspect were not feasible. Another limitation is that the study focused on peri domestic exposures, without extensively investigating extra-domestic activities - apart from occupational and agricultural activities but without detailing them. Many potentially relevant risk factors for leptospirosis-such as socioeconomic status, health literacy, prior disease knowledge, exposure to animals other than rodents, and local wastewater management-were scarcely or not assessed.

Finally, despite the methodological limitations of the study, the estimation of seroprevalence in understudied informal settlements and the identification of a local risk factor for exposure represent relevant findings for the local management of leptospirosis. Although their impact may be modest, these findings could support intervention campaigns around reported cases as well as broader awareness initiatives.

## Conclusion

This study confirmed a high rate of leptospirosis exposure factors but a rather limited seroprevalence among the surveyed population of three informal settlements of French Guiana. This may suggest that despite at-risk exposures the effective

transmission of leptospirosis remains limited within the study area because of individual protective measures or a limited contamination of environment. Broader serological surveys and environmental studies should clarify the areas of at-risk leptospirosis transmission in French Guiana.

## Supporting information

**S1 Text. Hantavirus Investigation in French Guiana 2022 – Questionnaire #1.**
(PDF)

**S2 Text. Hantavirus Investigation in French Guiana 2022 – Questionnaire #2 (complementary questionnaire administered during the delivery of results).**
(PDF)

**S1 Table. List of 17 variables used for computing the cumulative exposure score.**
(PDF)

**S1 Fig. Age distribution of participants.**
(PDF)

**S2 Fig. Distributions of the duration of living in French Guiana among study participants.**
(PDF)

**S3 Fig. Distribution of the duration of residence in the study area among study participants.**
(PDF)

**S4 Fig. Water standpipe in close proximity to garbage deposits (*Photo credit: Margot Oberlis, Mobile Environmental Health Team, French Red Cross, Cayenne, French Guiana*).**
(PDF)

## Acknowledgments

The authors would like to thank people among Agence Régional de Santé Guyane (Mathilde Ballet, Mathilde Hangard, Alban Lemonnier, Franky Mubenga, Adrien Ortelli, Solène Wiedner-Papin), Institut Pasteur de la Guyane (Damien Donato, Dominique Rousset), Centre National de Référence de la Leptospirose Institut Pasteur de Paris (Pascale Bourhy, Sofia Pombo, Farida Zinini, Céline Lorioux, Jean-François Mariet), La Croix-Rouge francaise (Karl Kpossou, Diana Perez-Aranguena, Luxamarre Saint Hervé, Marckenson Therasse) and Cayenne Hospital (Frédégonde About, Amadou Oury Baldé, Yoris Demars, Olivier Lesens, Camille Meunier, Marina Palomar, Déborah Porez, Mathilde Zenou), Santé Publique France Guyane (Luisiane Carvalho, Sophie Devos, Marion Guyot, Tiphanie Succo), Guyane Promo Santé (Jean-Luc Bauzat) and others who contributed to the Hantavirus survey and the ancillary study on leptospirosis. The authors acknowledge Prof. Maylis Douine (Cayenne University Hospital) for sharing the MAT data of ORPAL1 and ORPAL2 studies.

## Author contributions

**Conceptualization:** Paul Le Turnier, Margot Oberlis, Anne Lavergne, Loïc Epelboin.

**Data curation:** Margot Oberlis.

**Formal analysis:** Paul Le Turnier, Mathieu Picardeau.

**Investigation:** Paul Le Turnier, Margot Oberlis, Loïc Epelboin.

**Methodology:** Paul Le Turnier, Loïc Epelboin, Mathieu Picardeau.

**Project administration:** Anne Lavergne.

**Resources:** Anne Lavergne, Mathieu Picardeau.

**Supervision:** Margot Oberlis, Loïc Epelboin.

**Validation:** Mathieu Picardeau.

**Writing – original draft:** Paul Le Turnier.

**Writing – review & editing:** Margot Oberlis, Anne Lavergne, Loïc Epelboin, Mathieu Picardeau.

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
