## [Decision Letter · Decision Letter 0]

28 Oct 2025

Leptospirosis Seroprevalence and Exposure Factors in Informal Settlements of French Guiana: an Opportunistic Survey

Dear Dr. Le Turnier,

Thank you for submitting your manuscript to PLOS Neglected Tropical Diseases. After careful consideration, we feel that it has merit but does not fully meet PLOS Neglected Tropical Diseases's publication criteria as it currently stands. Therefore, we invite you to submit a revised version of the manuscript that addresses the points raised during the review process.

Please submit your revised manuscript within 60 days Dec 27 2025 11:59PM. If you will need more time than this to complete your revisions, please reply to this message or contact the journal office at plosntds@plos.org. Please include the following items when submitting your revised manuscript:

We look forward to receiving your revised manuscript.

Kind regards,

Joseph M. Vinetz

Section Editor

Shaden Kamhawi

co-Editor-in-Chief

Paul Brindley

co-Editor-in-Chief

**Additional Editor Comments:**

There are important limitations of this study that simply must be addressed in a revision. It is recognized that such studies are very hard to do in the resource limited setting. However, the limitations here are important, and the lack of systematic sampling must be addressed.  

**Journal Requirements:**

1) Please upload all main figures as separate Figure files in .tif or .eps format. For more information about how to convert and format your figure files please see our guidelines: 

2) We have noticed that you have uploaded Supporting Information files, but you have not included a list of legends. Please add a full list of legends for your Supporting Information files after the references list.

3) Some material included in your submission may be copyrighted. According to PLOSu2019s copyright policy, authors who use figures or other material (e.g., graphics, clipart, maps) from another author or copyright holder must demonstrate or obtain permission to publish this material under the Creative Commons Attribution 4.0 International (CC BY 4.0) License used by PLOS journals. Please closely review the details of PLOSu2019s copyright requirements here: PLOS Licenses and Copyright. If you need to request permissions from a copyright holder, you may use PLOS's Copyright Content Permission form.

Potential Copyright Issues:

i) Please confirm (a) that you are the photographer of S7, or (b) provide written permission from the photographer to publish the photo(s) under our CC BY 4.0 license.

4) We note that you have indicated that there are restrictions to data sharing for this study. PLOS only allows data to be available upon request if there are legal or ethical restrictions on sharing data publicly. For more information on unacceptable data access restrictions, please see https://journals.plos.org/plosntds/s/data-availability#loc-unacceptable-data-access-restrictions.

b) If there are no restrictions, please upload the minimal anonymized data set necessary to replicate your study findings to a stable, public repository and provide us with the relevant URLs, DOIs, or accession numbers. For a list of recommended repositories, please see https://journals.plos.org/plosone/s/recommended-repositories. You also have the option of uploading the data as Supporting Information files, but we would recommend depositing data directly to a data repository if possible.

5) Please provide a detailed Financial Disclosure statement. This is published with the article. It must therefore be completed in full sentences and contain the exact wording you wish to be published.

1) Please clarify all sources of financial support for your study. List the grants, grant numbers, and organizations that funded your study, including funding received from your institution. Please note that suppliers of material support, including research materials, should be recognized in the Acknowledgements section rather than in the Financial Disclosure

2) State the initials, alongside each funding source, of each author to receive each grant. For example: "This work was supported by the National Institutes of Health (####### to AM; ###### to CJ) and the National Science Foundation (###### to AM)."

3) State what role the funders took in the study. If the funders had no role in your study, please state: "The funders had no role in study design, data collection and analysis, decision to publish, or preparation of the manuscript."

4) If any authors received a salary from any of your funders, please state which authors and which funders..

6) Your current Financial Disclosure states, "The author(s) received no specific funding for this work.".

However, your funding information on the submission form indicates receiving fund from Santé Publique France (SPF). 

Please indicate by return email the full and correct funding information for your study and confirm the order in which funding contributions should appear. Please be sure to indicate whether the funders played any role in the study design, data collection and analysis, decision to publish, or preparation of the manuscript.

**Reviewers' Comments:**

Reviewer's Responses to Questions

**Key Review Criteria Required for Acceptance?**

**Methods**

-Are the objectives of the study clearly articulated with a clear testable hypothesis stated?

-Is the study design appropriate to address the stated objectives?

-Is the population clearly described and appropriate for the hypothesis being tested?

-Is the sample size sufficient to ensure adequate power to address the hypothesis being tested?

-Were correct statistical analysis used to support conclusions?

-Are there concerns about ethical or regulatory requirements being met?

Reviewer #1: The objectives of the study are clearly articulated, the design is appropriate, population well described,

Reviewer #2: Abstract : In the abstract, it is difficult to understand the difference between the primary study and the ancillary study. It may not be necessary to specify this in the abstract.

Introduction : Line 92 and following : Here you assert the link between leptospirosis and poverty in French Guiana, but that is the research question of your work. I would be more cautious in assumptions here.

Line 129 : A map showing the location of these cases and the neighborhoods where the study was conducted, if possible with a representation of informal settlement areas, would be useful.

Line 153 : aide ?

Reviewer #3: 1. Line 132: Please provide an overview of the original survey’s sampling methods rather than referring only to a prior publication. It is not possible to assess the implications of the results without additional sampling details.

2. Line 129: Clarify the number of informal settlements sampled. Based on the text (“Three areas near Cayenne were selected for on-site investigations: Boutilier [Remire-Montjoly] and PK13 and PK16 [Macouria]”), it appears that three areas within two informal settlements were included—please confirm.

3. Please summarize, in brief, the variables included in the cumulative exposure score to provide a general understanding of what factors were considered.

4. Clarify the languages used for survey administration (e.g., French only, or also Haitian Creole or others).

5. Line 159: The following statement needs clarification: “It was initially anticipated to use a threshold of 1:50 for positivity threshold. However, many sera were contaminated and the MAT results were too complicated to interpret with this threshold.”

Please elaborate on how contamination influenced interpretability and the rationale for raising the threshold to 1:100.

6. Line 106: Clarify the population to which the study objective refers in the statement:

“The primary objective of the study was therefore to assess the prevalence of prior Leptospira exposure in the population.”

Presumably this refers to residents of the three informal settlements, but this should be stated explicitly.

**Results**

-Does the analysis presented match the analysis plan?

-Are the results clearly and completely presented?

-Are the figures (Tables, Images) of sufficient quality for clarity?

Reviewer #1: Yes, Yes

Reviewer #2: It may be useful to specify that missing data were not considered in the calculation of odds ratios and p-values.

Reviewer #3: 1. Table 2: The reported N is 266, but the leptospirosis-specific questionnaire appears to have been administered to only 190 participants. Please clarify the denominator used for this table.

**Conclusions**

-Are the conclusions supported by the data presented?

-Are the limitations of analysis clearly described?

-Do the authors discuss how these data can be helpful to advance our understanding of the topic under study?

-Is public health relevance addressed?

Reviewer #1: Yes, Yes, Yes

Reviewer #2: Discussion:

Given the context of exposure widely shared by residents of these neighbourhoods, it would be interesting to supplement this study with seroepidemiological studies in nearby neighbourhoods with permanent housing.

Line 293 : It might be useful to specify whether these seroprevalence studies were conducted in response to the occurrence of cases.

As the population is Haitian and may have been exposed in their country of origin and in transit countries (Suriname in particular), it would be interesting to discuss this and cite any references in these countries.

Conclusion :

One cannot say that 7.5% is low; I would weigh it up.

Reviewer #3: The discussion should explicitly acknowledge or expand on the implication of the following limitations:

- Non-representative (convenience) sampling design

- High number of exclusions and/or missing data

- Limited capacity to generalize findings beyond the sampled communities

**Editorial and Data Presentation Modifications?**

Reviewer #1: The manuscript could benefit from some editing . A few examples are given below.

Material and Methods

Line 113-115: Health mediators helped the investigator and the participants when necessary during the on-field survey.

Please better specify the nature of ‘help’ provided. Did you mean to say: Health mediators assisted participants in filling out the questionnaire when necessary.

Discussion

Line 304-305: Because of a different titer threshold comparison was limited.

Consider: Since a different titer threshold was used, comparison may be questionable.

Line 342-343: Additionally, the time and place of acquisition of the previous exposure to leptospirosis cannot be ascertained since MAT remains positive for a long time.

Consider replacing ‘for a long time’ by ‘for several years’.

Reviewer #2: (No Response)

Reviewer #3: (No Response)

**Summary and General Comments**

Reviewer #1: The manuscript is adequate. The subject does not importantly add to novelty in science, but is OK. It is well written. My comments on editing are included in the section above.

Reviewer #2: Thank you for the opportunity to review this manuscript. The article deals with an important topic in a geographical region where it has been little studied and provides original findings that improve our knowledge and are likely to have practical implications. In particular, the estimation of seroprevalence in these understudied informal settlements and the identification of walking barefoot on wet ground as a risk factor for exposure are important findings that can be used for prevention purposes. Overall, the article is well written, based on a clear and high-quality methodology, and fits well with the journal's themes.

Reviewer #3: 1. General Overview

The authors report on Leptospira seropositivity, assessed by microscopic agglutination test (MAT) using a 17-serogroup panel, among what appears to be a convenience sample of 190 individuals from three informal settlements surrounding Cayenne, the capital of French Guiana, in 2022. The original serological sampling and survey tools were designed for a hantavirus investigation following an apparent increase in cases in preceding years. The present study repurposes those samples and metadata, supplemented with a leptospirosis-specific questionnaire administered to the same participants. The principal finding is a leptospirosis seroprevalence of 7.5% (95% CI: 4.7–11.4) using a ≥1:100 MAT threshold—lower than anticipated given the multiple exposure risks associated with informal settlement living conditions.

2. Major Comments

Representativeness and Generalizability: While I commend the authors for deriving additional insights from existing samples, the non-systematic nature of participant selection raises major concerns about the representativeness of the data. Given the low observed seroprevalence, even small changes in case numbers could substantially alter the prevalence estimate.

Public Health Relevance: It remains unclear how the current findings can meaningfully inform public health policy or programming, given the convenience sampling design and limited external validity. The authors should clarify what practical insights—if any—can be reasonably drawn from these results.

3. Additional Comments

Title: Please specify that the study was conducted in three informal settlements. The current title implies coverage of all informal settlements nationally. A clearer version might read:

Leptospirosis Seroprevalence and Exposure Factors in Three Informal Settlements of French Guiana: An Opportunistic Survey.

Line 93: The following sentence requires elaboration to better delineate how leptospirosis disproportionately affects vulnerable populations: “A third of affected individuals lacked health insurance and a third lived in informal settlements indicating that leptospirosis disproportionately affects vulnerable populations.” I'm a little confused by the logic of this sentence. Can the authors explicitly clarify how A (A third of affected individuals lacked health insurance) and B ( third lived in informal settlements) lead to C (leptospirosis disproportionately affects vulnerable populations)

Data availability: The current Data Availability statement appears internally inconsistent. Lines 394–395 state that data cannot be shared due to “ethical and legal restrictions,” yet lines 398–401 indicate that access to de-identified data may be granted upon reasonable request following approval from Santé Publique France and CNIL authorization. Further, the Data Availability section indicates that de-identified data can be accessed only after multiple institutional and CNIL approvals. Given that CNIL authorization governs identifiable data but does not restrict the sharing of fully anonymized datasets, it is unclear why the data cannot be made openly available in de-identified form. I encourage the authors to clarify whether these restrictions are institutional policy rather than legal requirement, and to consider providing an openly accessible anonymized dataset in alignment with PLOS NTD’s data-sharing standards.

PLOS authors have the option to publish the peer review history of their article (what does this mean? ). If published, this will include your full peer review and any attached files.

**Do you want your identity to be public for this peer review?** For information about this choice, including consent withdrawal, please see our Privacy Policy .

Reviewer #1: No

Reviewer #2: **Yes: ** Nicolas Vignier

Reviewer #3: No

**Figure resubmission:**
---

## [Editor Report · Decision Letter 1]

17 Nov 2025

Dear Dr. Le Turnier,

We are pleased to inform you that your manuscript 'Leptospirosis Seroprevalence and Exposure Factors in Three Informal Settlements of French Guiana: an Opportunistic Survey' has been provisionally accepted for publication in PLOS Neglected Tropical Diseases.

Best regards,

Joseph M. Vinetz

Section Editor

Joseph Vinetz

Section Editor

Shaden Kamhawi

co-Editor-in-Chief

Paul Brindley

co-Editor-in-Chief

---

## [Editor Report · Acceptance letter]

Dear Dr. Le Turnier,

We are delighted to inform you that your manuscript, "Leptospirosis Seroprevalence and Exposure Factors in Three Informal Settlements of French Guiana: an Opportunistic Survey," has been formally accepted for publication in PLOS Neglected Tropical Diseases.

Best regards,

Shaden Kamhawi

co-Editor-in-Chief

Paul Brindley

co-Editor-in-Chief
